# Spectral indices with different spatial resolutions in recognizing soybean phenology

**Airton Andrade da Silva**[1], **Francisco Charles dos Santos Silva**[1]*, **Claudinei Martins Guimarães**[2], **Ibrahim A. Saleh**[3], **José Francisco da Crus Neto**[1], **Mohamed A. El-Tayeb**[4], **Mostafa A. Abdel-Maksoud**[4], **Jorge González Aguilera**[5], **Hamada AbdElgawad**[6], **Alan Mario Zuffo**[1]*

1 Universidade Estadual do Maranhão, Balsas, Maranhão, Brazil, 2 Instituto Federal Goiano, Morrinhos, Goiano, Brazil, 3 Faculty of Science, Zarqa University, Zarqa, Jordan, 4 Department of Botany and Microbiology, College of Science, King Saud University, Riyadh, Saudi Arabia, 5 Universidade Estadual de Mato Grosso do Sul, Cassilandia, Mato Grosso do Sul, Brazil, 6 Integrated Molecular Plant Physiology Research, Department of Biology, University of Antwerp, Antwerp, Belgium

* franciscocharlessilva@professor.uema.br (FCSS); alan_zuffo@hotmail.com (AMZ)

**Data Availability Statement:** All relevant data is available in the GitHub repository at the following link: https://github.com/FSilva-826/DADOS---AMAZONIA1-CENTINEL2.

## Abstract

The aim of the present research was to evaluate the efficiency of different vegetation indices (VI) obtained from satellites with varying spatial resolutions in discriminating the phenological stages of soybean crops. The experiment was carried out in a soybean cultivation area irrigated by central pivot, in Balsas, MA, Brazil, where weekly assessments of phenology and leaf area index were carried out. Throughout the crop cycle, spectral data from the study area were collected from sensors, onboard the Sentinel-2 and Amazônia-1 satellites. The images obtained were processed to obtain the VI based on NIR (NDVI, NDWI and SAVI) and RGB (VARI, IV GREEN and GLI), for the different phenological stages of the crop. The efficiency in identifying phenological stages by VI was determined through discriminant analysis and the Algorithm Neural Network–ANN, where the best classifications presented an Apparent Error Rate (APER) equal to zero. The APER for the discriminant analysis varied between 53.4% and 70.4% while, for the ANN, it was between 47.4% and 73.9%, making it not possible to identify which of the two analysis techniques is more appropriate. The study results demonstrated that the difference in sensors spatial resolution is not a determining factor in the correct identification of soybean phenological stages. Although no VI, obtained from the Amazônia-1 and Sentinel-2 sensor systems, was 100% effective in identifying all phenological stages, specific indices can be used to identify some key phenological stages of soybean crops, such as: flowering ($R_1$ and $R_2$); pod development ($R_4$); grain development ($R_{5.1}$); and plant physiological maturity ($R_8$). Therefore, VI obtained from orbital sensors are effective in identifying soybean phenological stages quickly and cheaply.

## Introduction

Soybean [*Glicyne max* (L.) Merril] is a crop of high economic importance, due to the revenue generated from its numerous uses and by-products created; social importance, due to the

**Funding:** This study was funded by the College of Food and Agriculture Sciences, King Saud University, RSPD2024R678 (to Mohamed A. El-Tayeb). This study was also funded by a scholarship from CAPES, Coordination for the Improvement of Higher Education Personnel, 88887.677482/2022-00 (to Airton Andrade da Silva). The funders had no role in study design, data collection and analysis, decision to publish, or preparation of the manuscript.

**Competing interests:** NO authors have competing interests.

generation of employment resulting in regional development, and, mainly, for guaranteeing global food security, as it can be used in human and animal nutrition [1]. Therefore, research aimed at adding technologies and knowledge within the soy-bean production system, seeking to increase yield, is essential.

In this aspect, precision agriculture has become an ally in the process of optimizing soybean production, assisting in decision-making regarding agricultural practices based on the interpretation and integration of information from different data sources [2]. However, the aforementioned technique has presented deficiencies, in part, due to lesser efforts in incorporating the physiological principles of crop responses to environmental variation, such as crop phenology [3].

Soybean phenology is an important variable to be considered in decision-making during crop conjunction, such as choosing the sowing date, irrigation management decisions [4], lodging management [5], foliar application of nutrients [6], and/or phytosanitary management [7]. However, identifying the correct phenological stage in large areas and with different sowing dates becomes an expensive task, which demands time and technical knowledge.

Faced with this problem, vegetation indexes (VI) can be tools used to monitor soybean phenology during plant development, based on estimating the amount of electromagnetic energy reflected by the crop canopy. The technique can be used after interaction with pigments, water and intercellular spaces inside the leaf, providing data on the physiological state of the plant [8].

Among the potential advantages of using VI for monitoring soybean phenology is the speed in obtaining images, which are made available free of charge in catalogs and on mission websites, and the low demand for labor for the evaluation [9].

In recent years, different studies have demonstrated the efficiency of using of VI in differentiating the phenological stages of different crops, such as the normalized differentiation vegetation index (NDVI) used in soybean cultivation [10, 11]; the Normalized Differentiation Water Index (NDWI), for corn and soybeans [9]; the soil-adjusted vegetation index (SAVI), in wheat, corn and soybeans [12, 13]; the visible atmospheric resistance index (VARI), for corn and soybeans [14]; the visible green atmospheric resistance index (IV GREEN), in rice [15]; and the green leaf index (GLI), in corn [16].

The VARI, IV GREEN and GLI indices, as they contain only the red, green and blue bands in their composition, are classified as RGB indices. These consider wavelengths of the visible spectrum that relate only to leaf pigments (chlorophyll, carotenoids, anthocyanins and xanthophylls), which are responsible for absorbing 80–90% of visible light, with a peak of 0.55 μm, and reflect 10–20% of visible light, mainly the green band [17].

The NDVI, NDWI and SAVI indices have information from the near-infrared (NIR) range in their composition, being classified as NIR indices. This spectral range is very sensitive to the variation in vegetation biomass and, consequently, to the variation in plant growth and development [17].

Although there are many studies on the efficiency of different VIs in identifying the different phenological stages of soybeans and the evaluation of different sensor systems, with different spatial resolutions, are scarce, as the available studies use only one source of spectral data. Spatial resolution corresponds to the area on the ground represented in a single pixel of a digital image.

Given the above, the aim of the present research was to evaluate the efficiency of different vegetation indices (VI) obtained from satellites with varying spatial resolutions in discriminating the phenological stages of soybean crops.

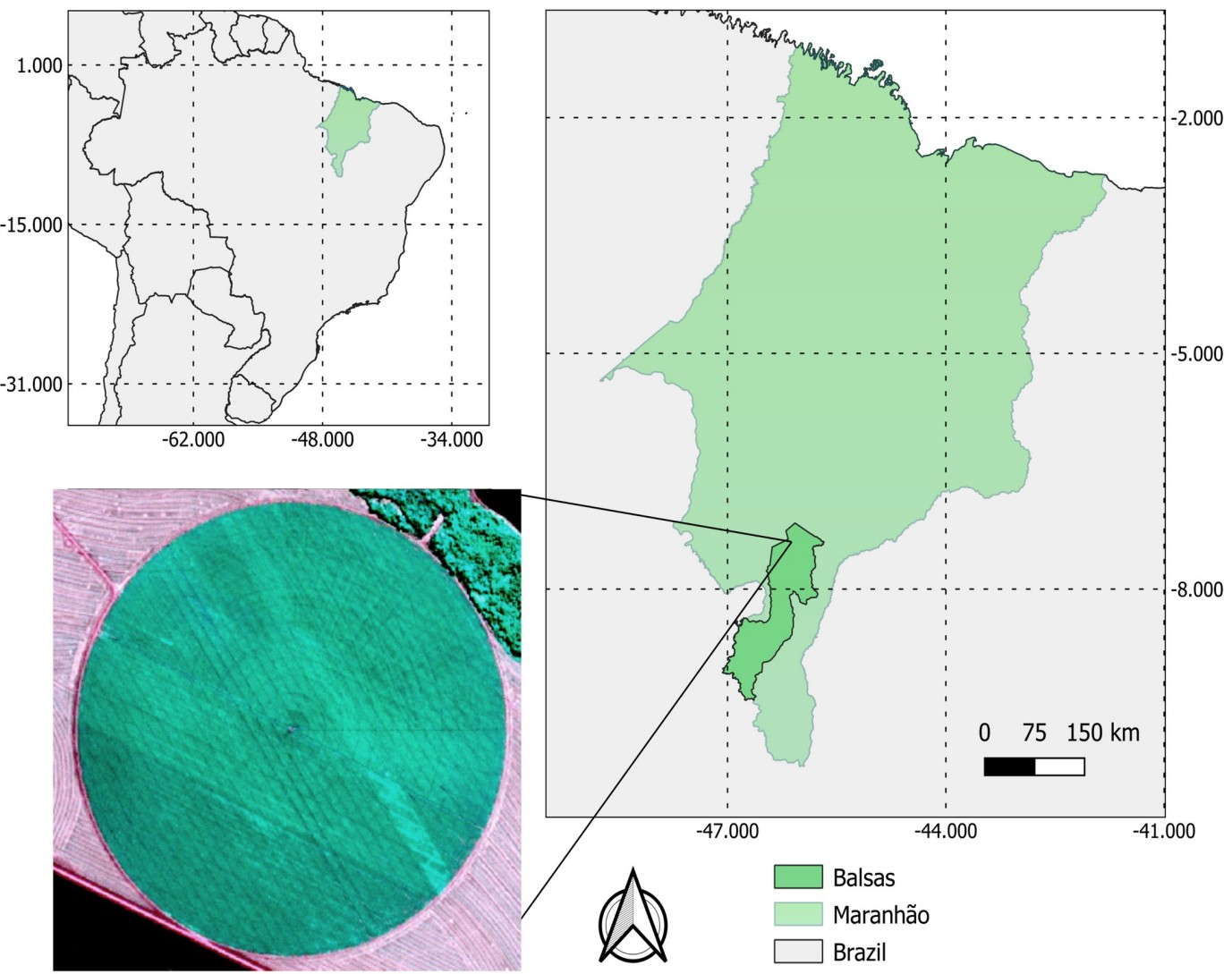

**Fig 1. Location of the experiment area.** Maps of the Maranhão State (A), Brazil (B) and the experimental area location (C). Source: [19].

## Material and methods

### Experiment location

The present research was carried out between April and July 2022, in a commercial production area of soybean seeds, with 80 ha, located 30 km from the municipality of Balsas, MA, Brazil (Fig 1), where the soybean, cultivar TMG2383, was conducted under central pivot irrigation. The location is at 7˚ 31' 59" South, 46˚ 2' 6" West, 243 meters altitude and has a rainy tropical climate (Aw) and an average temperature of 27.1 ˚C, according to Köppen climate classification. The average annual rainfall is 1175 mm, with the highest rainfall in the months of November to April, when they account for 85% of the total [18].

The soil in the experimental area was classified as Red Oxisol [20]. Meteorological data on average temperature, total precipitation and cloudiness for the study period (Fig 2) were obtained by the meteorological station of the Brazilian National Institute of Meteorology (INMET), present in the study region [21].

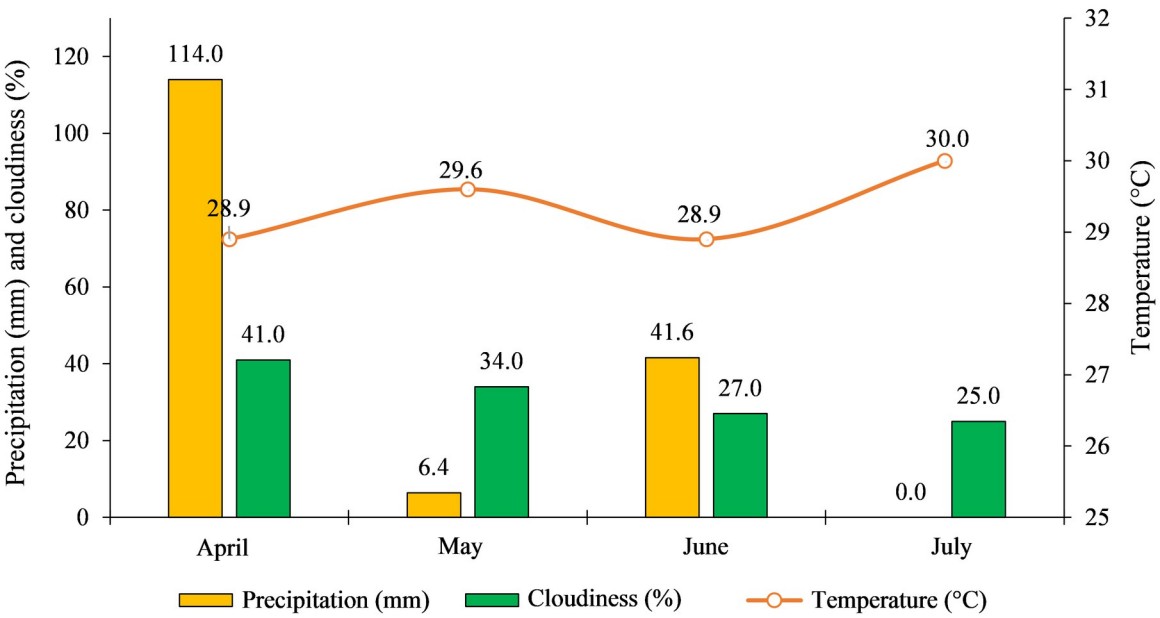

**Fig 2. Weather conditions during the experiment.** Total precipitation, cloudiness and average temperature, during the months which the experiment was carried out, in Balsas, MA, in the year 2022. Source: [21].

## Conducting the experiment

The experiment was carried out in a randomized block design, with four replications. Each measure point (replication) consisted of an area of 5 meters in radius, for Sentinel-2 products, and 30 meters in radius, for Amazonia-1 products, whose centers were demarcated with a global positioning system (GPS) navigation receiver.

After soybean germination, weekly monitoring of plant phenology was carried, according to [22] (Table 1) and also the leaf area index, using the dry disc method [23], adapted to a cylinder with a section of 7 cm2 and 10 discs per sample of three plants.

**Table 1. Soybean phenological stages.**

| STAGE | DESCRIPTION |
|---|---|
| $V_1$ | A pair of unifoliate leaves (or a node); |
| $V_2$ | First fully developed trefoil (or two nodes); |
| $V_3$ | Two fully developed trefoils (or three nodes); |
| $V_n$ | Vegetative stages until reaching the reproductive stage, from the emission of the first floral bud; |
| R1 | An open flower at any node on the main stem; |
| $R_2$ | An open flower on one of the two upper nodes of the main stem, with a fully developed leaf; |
| R3 | Pod 0.5 cm to 2.0 cm in one of the four upper nodes of the main stem; |
| $R_4$ | Pod fully developed (> 2.0 cm) on one of the four upper nodes of the main stem; |
| $R_{5.1}$, $R_{5.2}$, $R_{5.3}$, $R_{5.4}$ e $R_{5.5}$ | Beginning of grain filling (<10% to 100% grain) in one of the four upper nodes of the main stem; |
| $R_6$ | Full or complete grain in one of the four upper nodes of the main stem; |
| $R_7$ | Beginning of maturation: a pod with a mature color on the main stem; |
| $R_8$ | Full maturity (harvest): more than 95% of the pods are ripe in color. |

Source: [22]

**Table 2. Technical specifications of the MSI sensor, onboard the Sentinel-2 satellite, and the WFI sensor, onboard the Amazonia-2 satellite.**

| Satellite | Temporal Resolution | Spectral Resolution (μm) | Radiometric Resolution | Spatial Resolution |
|---|---|---|---|---|
| Sentinel-2 | 5 days* | Blue (0.49–0.56) | 10 bits | 10 m |
| | | Green (0.56–0.67) | | |
| | | Red (0.67–0.84) | | |
| | | Near Infrared (0.84–0.87) | | |
| Amazonia-1 | 5 days | Blue (0.45–0.52) | 10 bits | ~ 65 m |
| | | Green (0.52–0.59) | | |
| | | Red (0.63–0.69) | | |
| | | Near Infrared (0.77–0.89) | | |

* Resolution of constellation with Multispectral Imager (MSI) sensor

Images collected by the Multispectral Imager (MSI) sensor, onboard the Sentinel-2 satellite, were used as a source of reflectance data, which were obtained from the image catalog of the Copernicus Open Access Center (https://scihub.copernicus.eu/dhus/#/home), at L2A processing level. The L2A images were corrected for clouds and cloud shadows, aerosol optical thickness, water vapor, orthorectified surface reflectance with multispectral, and subpixel multitemporal registration accuracy. The sensor provided products with a temporal resolution of 5 days, spatial resolution, which varies from 10 m to 65 m, radiometric resolution of 10 bits and spectral resolution of 13 bands, however, in the present study, only bands with a spatial resolution of 10 were used. meters (Table 2).

Images from the WFI (Wide Field Imager) orbital sensor, onboard the Amazonia-1 satellite, were also used as a source of reflectance data, which were obtained from the image catalog of INPE—National Institute for Space Research (http://www2.dgi.inpe.br/catalogo/explore). The sensor provided products with resolution in accordance with Table 2.

Both satellites were chosen due to their temporal resolution of 5 days, which allows better monitoring of phenology throughout the crop cycle, and spatial resolution of 10 and 64 m, enabling comparison of data quality. Because, according to [24], monitoring phenology, using orbital sensors, must have the greatest amount of data during the cycle, favoring the monitoring of the different stages of the crop.

The temporal resolution of the sensors allowed the collection of 14 images for the Sentinel-2 satellite and 11 images for the Amazonia-1 satellite, throughout the crop cycle, which made it possible to obtain spectral information for the phenological stages from $V_3$ to $R_8$ (Fig 3). The criteria for selecting images followed the availability for download of the respective image catalogs and that they did not present total cloud coverage in the study area.

## Image processing

The images obtained were processed in Qgis software version 3.26.3 [25] using Datum Sirgas 2000 UTM Zone 23S as the coordinate system. The images were submitted to the VI equations (Table 3), generating the products presented in Figs 4 and 5.

## Statistical analyzes

With the help of the GENES statistical program [31], the data obtained were subjected to the F test of the analysis of variance (ANOVA), which sought to verify the existence of a significant

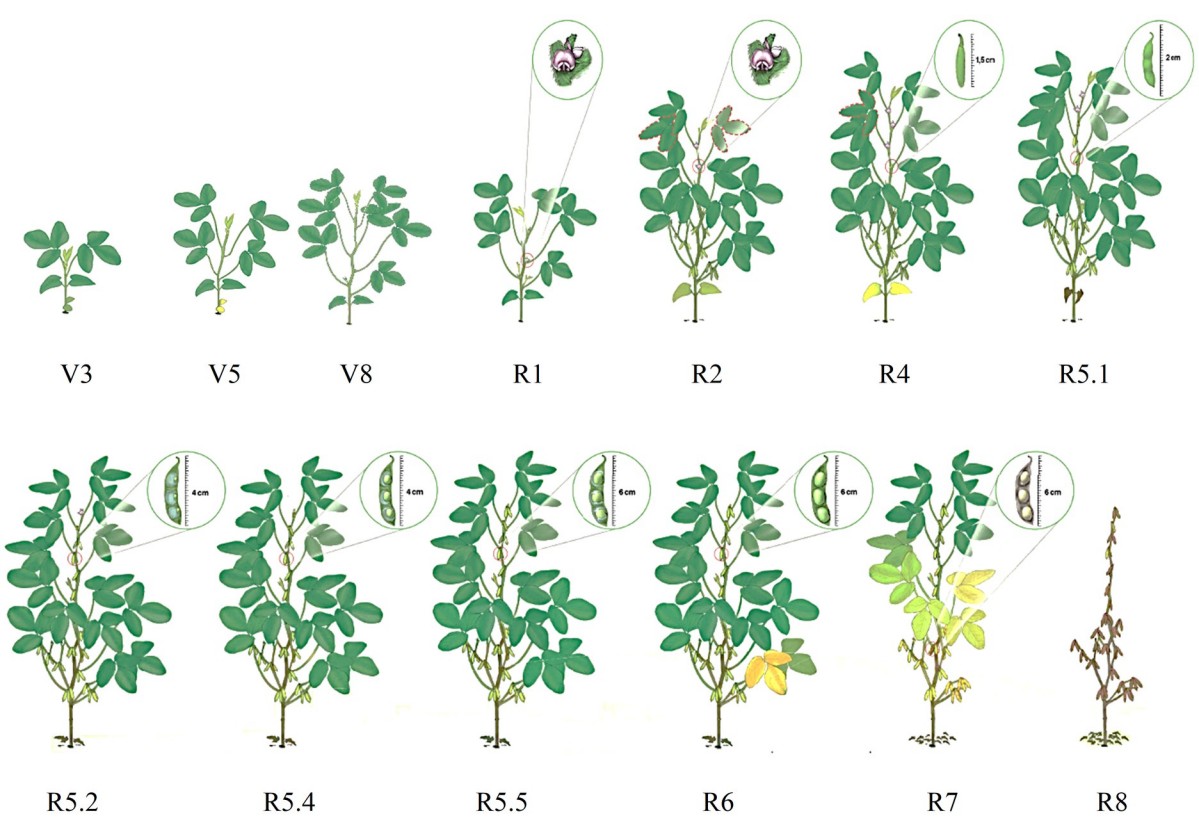

**Fig 3. Phenological stages of soybean crop.** Morphological aspect of soybeans at different phenological stages, from which spectral information was obtained by the Sentinel-2 and Amazonia-1 satellites. Source: Adapted from [22].

difference at 1% and 5% probability between the repetitions and between the phenological stages, throughout the cycle. of the crop, for the different vegetation indices (VIs).

In order to quantify the efficiency of each VI in differentiating the phenological stages of soybean, the values of the IVs that showed differences between the phenological stages were subjected to discriminant analysis by [32] and an Artificial Neural Network (ANN) algorithm type Multilayer Perceptron [33], with the aid of Genes and Weka 3.8.5 software (Waikato, New Zealand), respectively. The objective was to allow the algorithms to distinguish the clusters that corresponded to each of the phenological stages, based on the field spectral data associated with each of them.

Discriminant analysis was initially addressed by [34], and consists of obtaining mathematical functions capable of classifying a sample into one of several groups based on characteristics of these groups, seeking to minimize the probability of misclassification, that is, minimizing the probability of classifying mistakenly an individual in one population, when it actually belongs to another population. This analysis is a multivariate statistical technique used to discriminate and classify objects (Eq 1).

$$D_i(\tilde{x}) = \tilde{L}_i . \tilde{x} - \frac{1}{2} . \tilde{L}_i . \tilde{\mu}_i + ln(p_i) \qquad (1)$$

Where: $D_i(\tilde{x})$= population discriminant function 'i' of the random vector $\tilde{x}$; $\tilde{L}_i$ = population discriminant random vector 'i'; $\tilde{x}$= random feature vector; $\tilde{\mu}_i$= vector of population means 'i'; $p_i$ = population probability of occurrence 'i'.

**Table 3. Vegetation indices (VIs) used in the present research.**

| Type | Vegetation Indices (Vis) | Name | Equation | Reference |
|---|---|---|---|---|
| NIR | NDVI | Normalized Differentiation Vegetation Index | (B8-B4)/(B8+B4) | [26] |
| | NDWI | Normalized Differentiation Water Index | (B3-B8)/(B3+B8) | [27] |
| | SAVI | Soil-Adjusted Vegetation Index | (1+L)*[(B8•B4)÷(B8+B4+L)] | [28] |
| RGB | VARI | Visible Atmospheric Resistance Index | (B3-B4)/(B3+B4-B2) | [29] |
| | GLI | Green Leaf Index | (2*B3-B4-B2) / (2*B3+B4+B2) | [30] |
| | IV GREEN | Atmospheric resistance indices Visible Green | (B3-B4)/(B3+B4) | [29] |

Notes: B4: red reflectance, B8: near infrared reflectance, B3: green reflectance; B2: blue reflectance, L = adjustment factor that can vary from 0 to 1.

Before applying the discriminant analysis, the data were subjected to the analysis of normality, using the Kolmogorov-Smirnov test (having presented normality), of identification of outliers through box plot graphs (these being removed when identified), as well as the Box's M test, to check the similarity of the dispersion matrices of the independent variables.

In turn, ANN was adopted because the pattern of variation in VIs throughout the soybean development cycle required non-linear statistical modeling. The ANN was tested using the standard architecture of the Weka 3.8.5 software, consisting of a Multilayer Perceptron, with a single hidden layer, formed by a number of neurons equal to the sum of the number of variables analyzed with the number of classes, divided by two, and using cross-validation with k-fold = 10.

The efficiency of the discriminant functions in correctly classifying each phenological stage was estimated by the apparent error rate (Eq 2), where higher values indicate lower accuracy in the differentiation.

$$APER = \frac{1}{N} \cdot \sum_{i=l}^{n} m_i \qquad (2)$$

Where: APER = apparent error rate; mi = number of wrong observations in groups; N = total number of ratings.

In the present study, only the VI that presented an apparent error rate equal to zero were considered efficient, that is, they allowed the correct identification of a stadium in 100% of the samples.

## Methodology summary

A summary of the methodology used in this research is illustrated in Fig 6.

## Results

All vegetations indices (VIs) behaved in accordance with the standard re-ported in the literature [35–37], demonstrating the quality of the spectral information obtained (Fig 7).

All VI analyzed, in both sensor systems (Sentinel-2 and Amazonia 1), showed significant differences between the phenological stages, at 5% probability, by the F test of the analysis of variance (Fig 7), making it possible to quantify the efficiency of distinguishing the phenological stages of the soybean crop for each VI, via discriminant analysis.

For the Sentinel-2 sensor system, the discriminant analysis for the NDVI, NDWI and SAVI indices showed the highest apparent error rates (APER), with 62.5, 60.7 and 62.5%, respectively, or that is, less than 50% efficiency in identifying the correct phenological stages. The IV

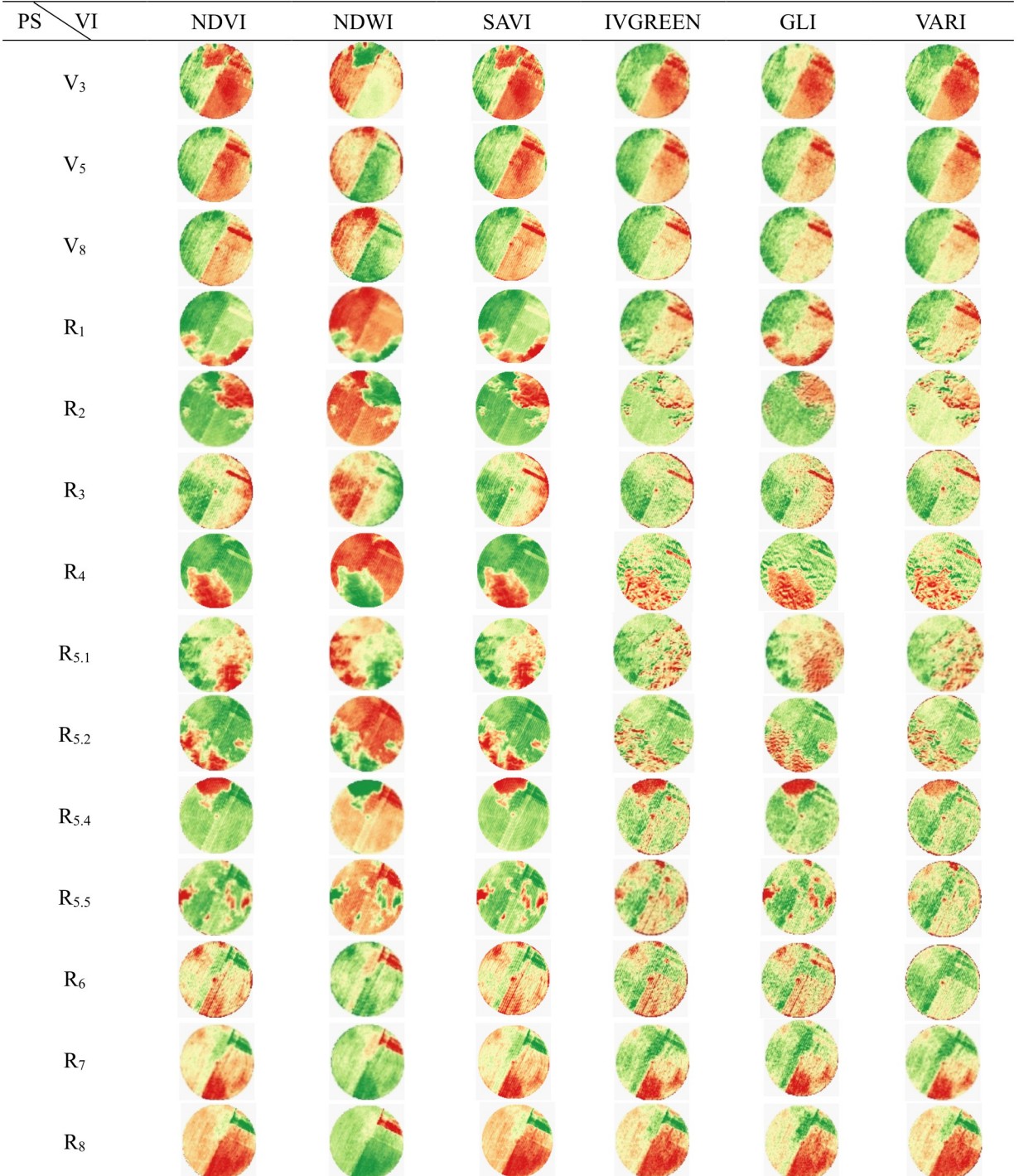

**Fig 4. Behavior of vegetation indices (VIs) throughout the soybean cycle, for Senti-nel-2 products, and for each phenological stage (PS) of the crop.**

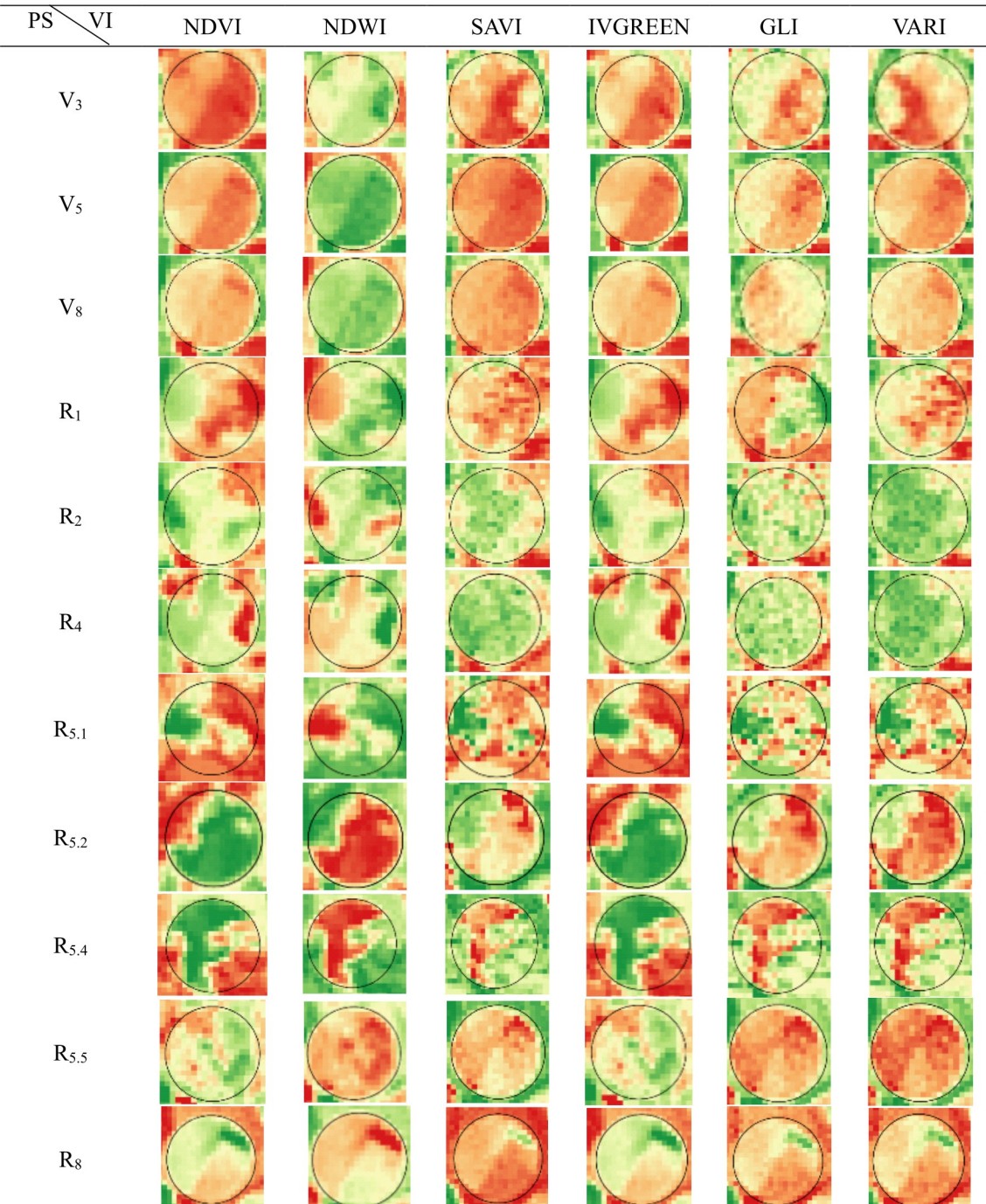

**Fig 5. Behavior of vegetation indices (IVs) throughout the soybean cycle, for Ama-zonia-1 products, and for each phenological stage (PS) of the crop.**

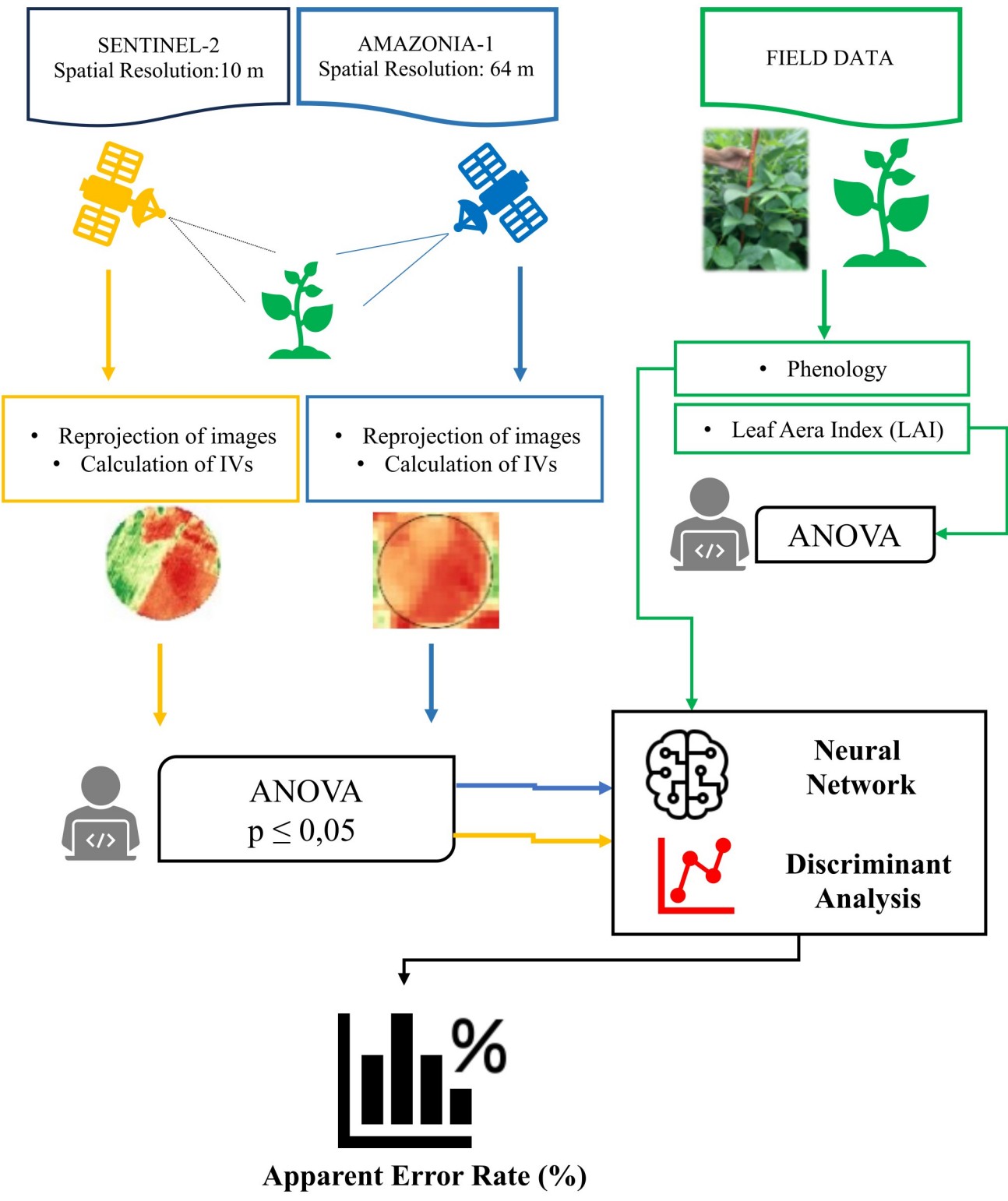

**Fig 6. Flowchart of the present research methodology.**

### Sentinel-2

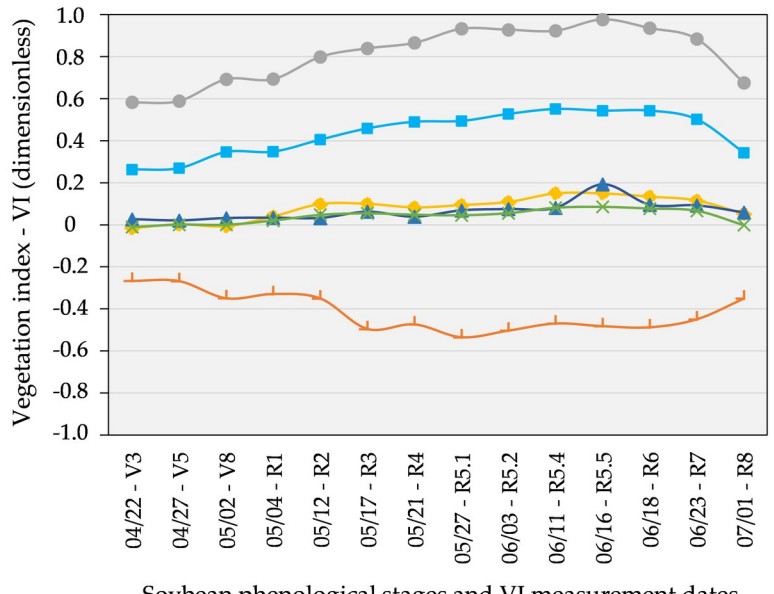

| | F Test |
|---|---|
| NDVI | Repetitions: 1.17 ns<br>Phenological stages: 78.74 ** |
| NDWI | Repetitions: 0.69 ns<br>Phenological stages: 4.89 ** |
| SAVI | Repetitions: 1.07 ns<br>Phenological stages: 79.98 ** |
| VARI | Repetitions: 2.55 *<br>Phenological stages: 189.27 ** |
| GLI | Repetitions: 1.38 ns<br>Phenological stages: 155.28 ** |
| IV GREEN | Repetitions: 0.54 ns<br>Phenological stages: 97.34 ** |

### Amazonia-1

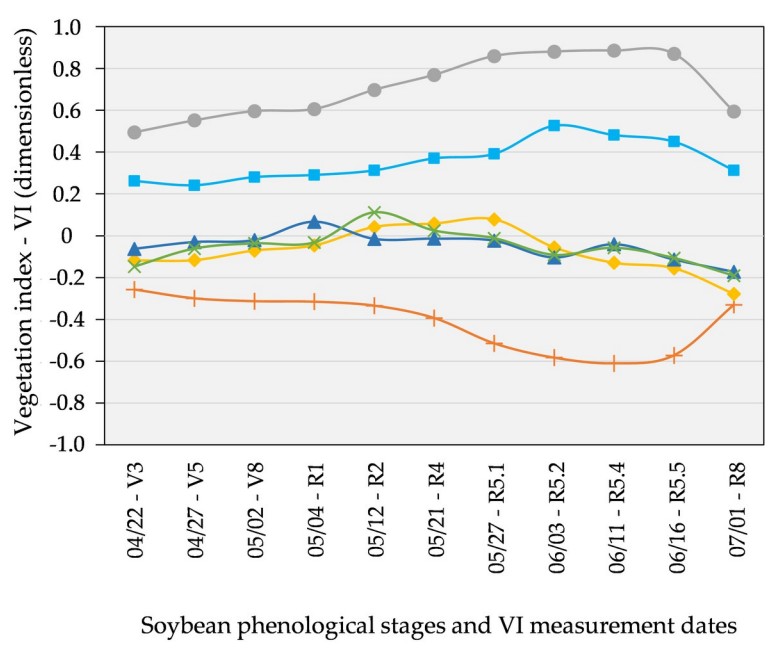

| | F Test |
|---|---|
| NDVI | Repetitions: 1.18 ns<br>Phenological stages: 6.22 ** |
| NDWI | Repetitions: 0.21 ns<br>Phenological stages: 106.47 ** |
| SAVI | Repetitions: 1.15 ns<br>Phenological stages: 34.75 ** |
| VARI | Repetitions: 1.61 ns<br>Phenological stages: 82.25 ** |
| GLI | Repetitions: 0.92 ns<br>Phenological stages: 28.85 ** |
| IV GREEN | Repetitions: 2.58 ns<br>Phenological stages: 13.17 ** |

NDVI — NDWI — SAVI — VARI — GLI — IVGREEN

**Fig 7. Variation of NIR and RGB vegetation indices from the Sentinel-2 and Amazonia-1 satellites.** Vegetation Indices NDVI, NDWI, SAVI (NIR type), and VARI, GLI and IV GREEN (RGB type). Notes: ns = not significant; ** = significant at 1% by the F test from ANOVA.

**Table 4. Apparent Error Rate (APER, %) of vegetation indices (VIs) for the Sentinel-2 and Amazonia-1 sensor systems, obtained via Anderson Discriminant Functions and Multilayer Perceptron Neural Network.**

| Satellite | Anderson Discriminant Functions | | Multilayer Perceptron Neural Network | |
|---|---|---|---|---|
| | **NIR** | **RGB** | **NIR** | **RGB** |
| Sentinel-2 | NDVI = 62.5 | VARI = 59.8 | NDVI = 51.8 | VARI = 63.4 |
| | NDWI = 60.7 | GLI = 58.9 | NDWI = 47.4 | GLI = 67.9 |
| | SAVI = 62.5 | IV GREEN = 56.2 | SAVI = 65.2 | IV GREEN = 57.1 |
| Amazonia-1 | NDVI = 70.4 | VARI = 63.6 | NDVI = 65.9 | VARI = 73.9 |
| | NDWI = 53.4 | GLI = 60.2 | NDWI = 60.2 | GLI = 58.0 |
| | SAVI = 57.9 | IV GREEN = 64.7 | SAVI = 53.4 | IV GREEN = 72.8 |

GREEN index provided the best efficiency in correctly identifying phenological stages, with an APER of only (56.2%). On the other hand, using the ANN algorithm, the NDWI index was the most efficient (47.4%) for APER (Table 4).

For the Amazonia-1 sensor system products, the discriminant analysis for NDVI, IV GREEN, VARI, GLI and SAVI showed the highest APER, 70.4, 64.7, 63.6, 60.2 and 57.9%, respectively, being the lowest for the NDWI index (53.4%) (Table 4).

The APER of the discriminant analysis varied between 53.4% and 70.4% while the APER of the ANN was between 47.4% and 73.9%. Therefore, it was not possible to identify which of the two analysis techniques (discriminant analysis or ANN) is more appropriate for processing spectral data aiming to identify soybean phenological stages, since the aforementioned techniques present variation in APER depending on the satellite or VI to be considered, and the combination between them (Table 4).

No VI obtained from the Amazonia-1 and Senti-nel-2 sensor systems was 100% effective in identifying the phenological stages of soybean crops, which demonstrates the impossibility of using these indices to simultaneously identify all stages of culture. On the other hand, some indices showed efficiency of 100% in identifying specific phenological stages (Figs 8 and 9).

No VI proved to be statistically efficient in identifying the vegetative stages of soybean development, with the most efficient ones showing a maximum effectiveness of 87.5% (Figs 8 and 9).

The correct classification of the stages that mark the flowering of the soybean crop, R1 (beginning of formation) and R2 (full flowering) was effective based on data from the Senti-nel-2 and Amazonia-1 Sensor System (Fig 8). The $R_1$ stage was successfully identified by the Amazonia-1 SAVI index, when submitted to the ANN Algorithm (Fig 9). The $R_2$ stage was correctly identified by the NIR-based indices (NDVI and SAVI) of Sentinel-2 and the RGB-type indices (IV Green and VARI) of the Amazonia-1 sensor system, when Discriminant Analysis was applied (Fig 8). The NDWI index, obtained from the Sentinel-2 satellite, was also effective in identifying the $R_2$ stage, when analyzed by the ANN Algorithm (Fig 9).

As for the reproductive stages, which mark the development of the pods, only the $R_4$ stage (fully formed pod) was successfully identified by the Amazonia-1 sensor, using the IV GREEN index, when analyzed by Discriminant Analysis (Fig 8).

The $R_{5.1}$ stage, which characterizes the development of grains inside the pods, was precisely identified using spectral information from the Sentinel-2 satellite, using the NDWI index, analyzed by Discriminant Analysis (Fig 8). The aforementioned stage ($R_{5.1}$) was also identified by the NDWI and SAVI indices, obtained based on data from Amazonia-1, when analyzed via the ANN Algorithm or Discriminant Analysis (Figs 8 and 9).

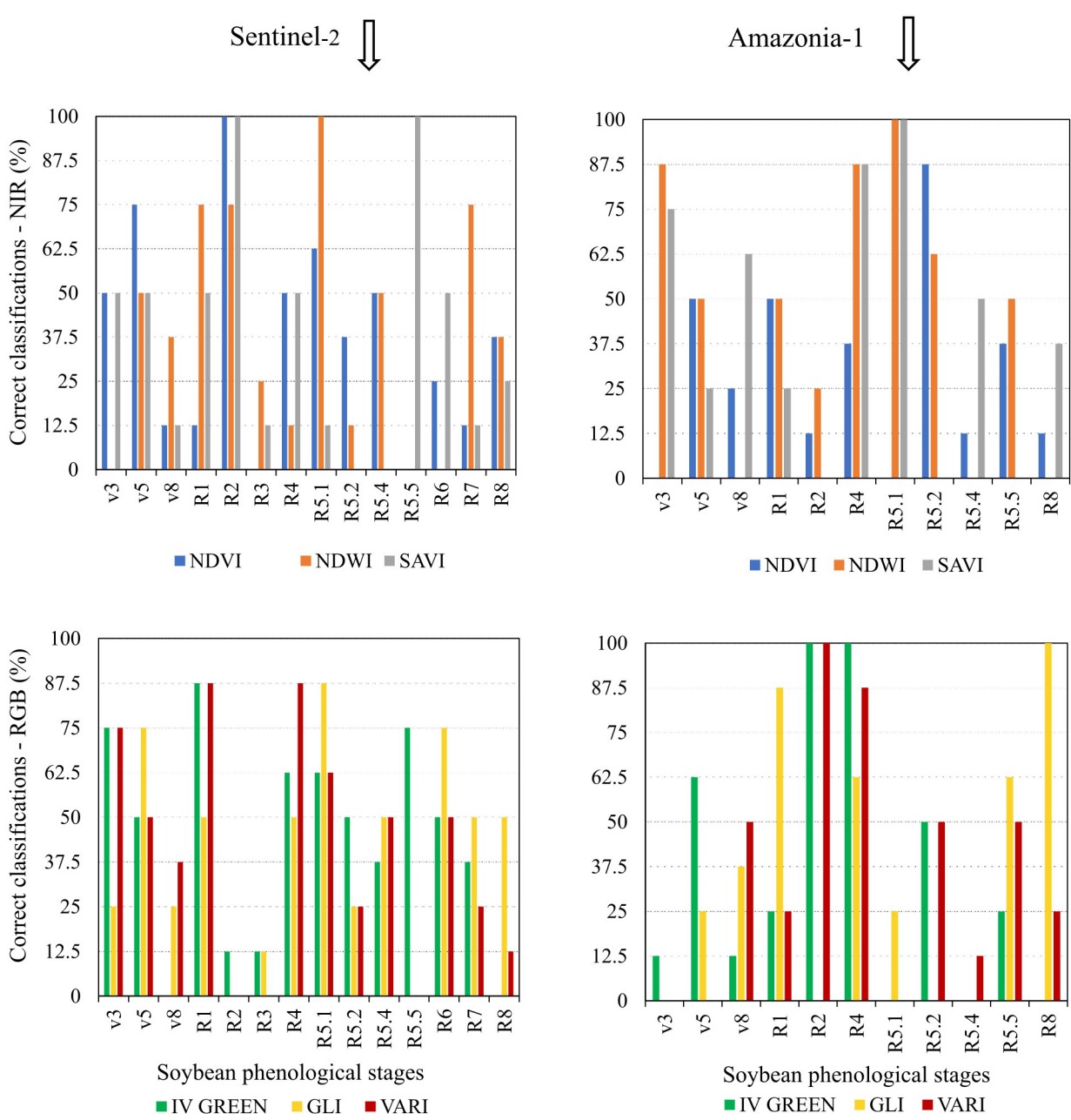

**Fig 8. Correct classifications of soybean phenological stages using different vegetation indices via discriminant analysis.** Vegetation Indices NDVI, NDWI, SAVI (NIR type), and VARI, GLI and IV GREEN (RGB type) from the Sentinel-2 and Amazonia-1 satellites.

In turn, the $R_8$ stage (full maturity of the soybean crop) was correctly identified only by the GLI index, obtained from spectral data from the Amazonia-1 satellite, regardless of the analysis technique, ANN Algorithm or Discriminant Analysis (Figs 8 and 9).

Regarding the performance of the statistical techniques used, despite the ability of Multi-layer Perceptron Neural Network algorithms (ANN) to identify non-linear patterns in a set of data, it was not possible to verify the superiority of this technique over Discriminant Analysis, once both of which have similar efficacy.

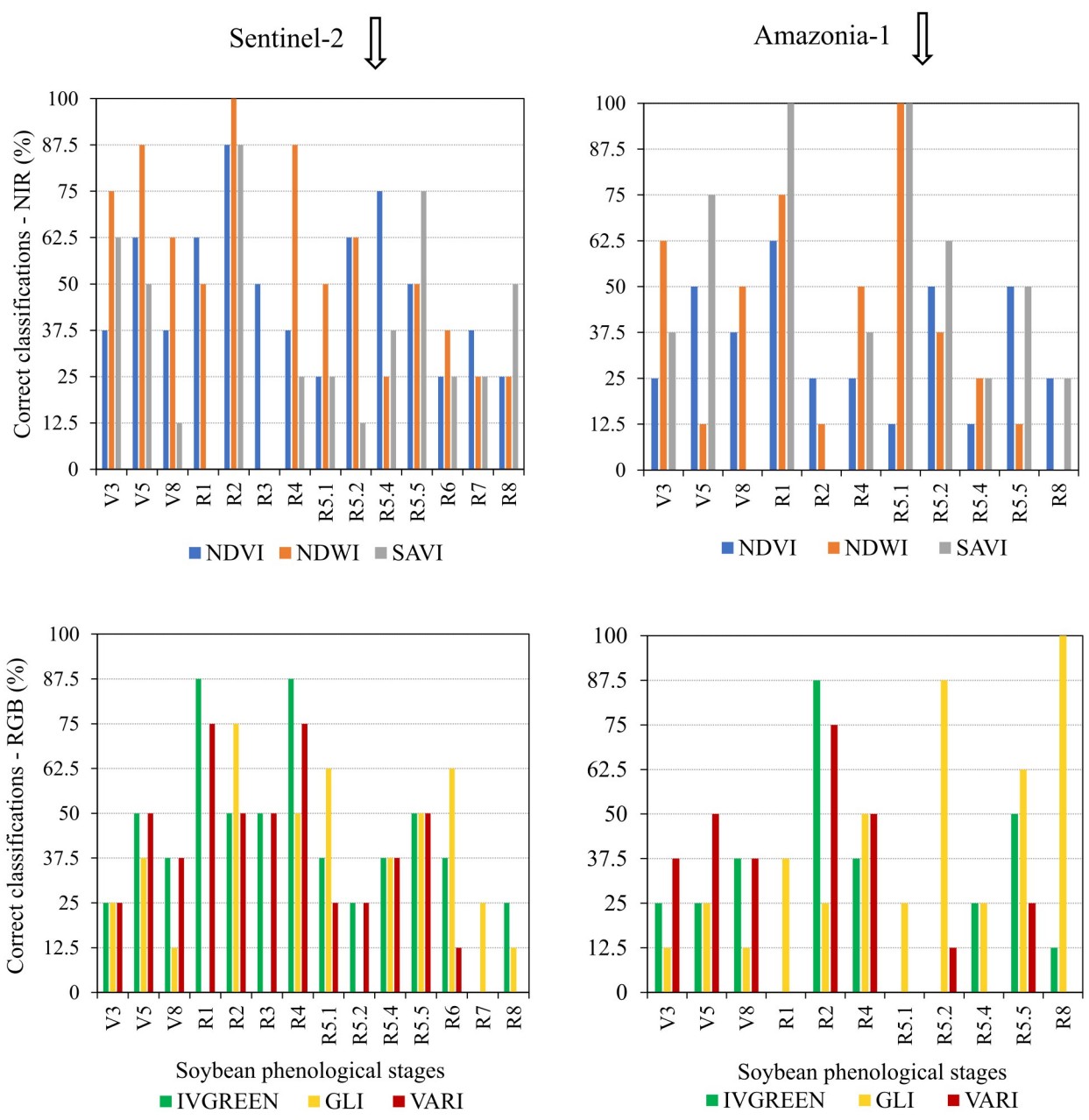

**Fig 9. Correct classifications of soybean phenological stages using different vegetation indices via multilayer perceptron neural network.** Vegetation Indices NDVI, NDWI, SAVI (NIR type), and VARI, GLI and IV GREEN (RGB type) from the Sentinel-2 and Amazonia-1 satellites.

## Discussion

The quality of the spectral information obtained (Fig 7) was confirmed because the behavior of all vegetations indices (VIs) is in accordance with the standard reported in the literature [36–39].

Several authors report that the spectral behavior of soybeans throughout the cycle presents low values at the beginning of the cycle, gradually increases to a maximum biomass and decreases with the end of the crop cycle [10, 12–16, 39]. The NDWI index showed behavior

similar to that mentioned above, however the curve was inverse due to its values being negative, as reported by [9]. The significant differences between the phenological stages for all VI analyzed (Fig 7), in both sensor systems (Sentinel-2 and Amazonia 1), made it possible to quantify the efficiency of distinguishing the phenological stages of the soybean crop for each VI, via discriminant analysis. [40], were able to classify different characteristics to differentiate soybean cultivars, based on the discriminant function, with a lower apparent error rate.

Although no VI obtained from the Amazonia-1 and Sentinel-2 sensor systems was 100% effective in simultaneously identifying all phenological stages of the soybean crop. Some indices made it possible to correctly identify stages $R_1$, $R_2$, $R_4$, $R_{5.1}$ and $R_8$.

In turn, the precise identification of stages $R_1$ and $R_2$ is essential for determining the appropriate time to apply nutritional and phytosanitary management measures and productivity prediction. For example, the $R_1$ stage of soybean cultivars with determinate growth, and $R_2$, for cultivars with indeterminate growth, are ideal for evaluating the nutritional status of the plant, by foliar diagnosis [41]. At these stages ($R_1$ and $R_2$), pesticide applications are also recommended to control white mold (*Sclerotinia sclerotiorum*), Asian rust (*Phakopsora pachyrhizi*), Brown spot (*Septoria glycines*), Powdery mildew (*Erysiphe diffusa*), Black spot target (*Corynespora cassiicola*) and Anthracnose (*Colletotrichum truncatum*) [42]. [43] defined the $R_1$ and $R_2$ stages as the best moment for predicting productivity in soybean crops, based on the Normalized Difference Vegetation Index (NDVI).

As for the $R_4$ stage, [44] defined it as the best time for predicting productivity in soybean crops based on the Soil Adjusted Vegetation Index (SAVI). At this stage, which marks the most critical beginning of development in terms of yield, supplemental irrigation may be a recommended practice with the aim of reducing abortion [45].

The correct identification of phenological stage $R_{5.1}$ is important because they are characterized by maximum nutrient and water requirements. Also, supplementary irrigation management is recommended in case of water deficit, favoring the absorption of nutrients from the soil, in addition to enriching seeds with molybdenum and predicting productivity, based on the Visible Atmospheric Resistance Index [45–47].

The $R_8$ stage must be correctly identified as it represents an ideal time for applying desiccants, favoring water loss of up to 13–14% of grain moisture, which provides an adequate soybean harvest [46].

The VIs discussed in this study fall into two categories: RGB in-dexes and indices that are based on the near-infrared (NIR). The VARI, IV GREEN and GLI indices, which are composed only of the blue, green and red bands, are categorized as RGB indices, and are associated only with leaf pigments (chlorophyll, carotenoids, anthocyanins and xanthophylls). On the other hand, the NDVI, NDWI and SAVI in-dexes include information from the near-infrared (NIR) range, which is highly sensitive to variation in plant biomass and, consequently, to variation in plant growth and development [17]. The relationship of these indices with morphophysiological aspects of plants explains their ability to identify changes in the previously mentioned soybean phenological stages.

The sensor systems studied showed similar effectiveness in identifying soybean phenological stages, both being capable of identifying stages $R_2$, $R_4$ and $R_{5.1}$. However, the Amazonia-1 satellite showed superiority with the precise identification of the $V_3$ and $R_8$ stages, demonstrating that the spatial resolution of 64 meters, of the Wide Field Imager (WFI) sensor embargoed on this satellite, is not a limiting factor for the identification of some phenological stages of soybeans, at least in large cultivation areas, as was the case in this study. However, studies are needed to investigate the efficiency of this sensor system in identifying soybean phenological stages with variation in the imagined cultivation area.

[48] managed to differentiate the initial stages of growth of soybeans and corn, using the spatial resolutions of 10 and 30 m of the MSI sensor on the Sentinel-2 satellite, demonstrating that different spatial resolutions have similar effectiveness for monitoring phenology in soybeans.

Regarding the types of VIs for the reproductive stages $R_1$, $R_2$, $R_4$ and $R_{5.1}$, the RGB indices were as efficient as the indices that have the NIR band.

RGB indices such as VARI, VIGREEN and GLI have received a lot of attention due to the possibility of their application based on data collected by simple and low-cost cameras that can be mounted on unmanned aerial vehicles (UAVs) [49]. Thus, the results of this research open the possibility of using RGB sensors in UAVs to differentiate the reproductive stages of soybeans with greater practicality and agility, as has already been observed by [50].

[51, 52] e [53] observed that RGB-type indices, compared to indices that use the infrared band, demonstrated greater yield for the concise prediction of grain yield, concentration of nitrogen and phosphorus in the leaf, proportion of carbon to nitrogen, under a wide range of nitrogen fertilization levels, and effects of phosphate fertilizer applications, in corn cultivation, in addition to potential use for selection of soybean cultivars with drought tolerance.

Another important point observed in the present study is that, regardless of the type of sensor system or type of vegetation index, the highest rates of effectiveness in classifying phenological stages were obtained for the reproductive stages of $R_1$, $R_2$ (flowering), $R_4$ (developed pod), $R_{5.1}$ (grain development). The possible explanation for this result lies in the variation in the magnitude of VI values throughout the crop cycle.

The spectral behavior of soybeans throughout the cycle presents low values at the beginning of the cycle, gradually increases and decreases as the end of the crop cycle approaches (Fig 7), corroborating [10, 12–16, 39]. The same RGB indices, which present a smaller magnitude of variation in their values throughout the crop cycle, also presented the same pattern of variation as observed for the VARI index of the Sentinel-2 satellite (Fig 7).

This aforementioned spectral behavior results in the proximity and even coincidence of the values of the VIs between the initial and final stages, as well as the similarity between the values of the indices in the final reproductive stages, $R_6$ to $R_8$. This similarity of values makes it impossible to effectively distinguish between the initial phenological stages and the final stages of crop development. This leaves the reproductive stages from $R_1$ to $R_{5.1}$, with more divergent values for the indices than the other stages, which results in higher efficiency rates in identifying these phenological stages. Similar results were obtained for rice by [54] using machine learning to identify the phenological stages of the crop.

To corroborate these results, the same pattern of variation is observed for the leaf area index—LAI (Fig 10). The LAI is defined as the relationship between the leaf area of a plant and the soil area occupied by it, with a high correlation between spectral in-dexes [22, 55].

Several authors, when studying the spectral behavior of soybeans, corn, wheat and rice, using RGB and Infrared vegetation indices, observed little variation in the values of the indices during the reproductive stages [9–16].

Although no VI was efficient in identifying all phenological stages of soybean crops, the indices were effective in identifying specific stages, demonstrating the possibility of using the technique in managing soybean crops. Furthermore, the free availability of reflectance data as well as processing software makes the non-use of this method of monitoring the development of soybean crops unjustified.

However, new studies that encompass other VI, as well as the application of combinations and/or simultaneous use of vegetation indexes to refine the identification of soybean phenological stages via remote sensing are necessary.

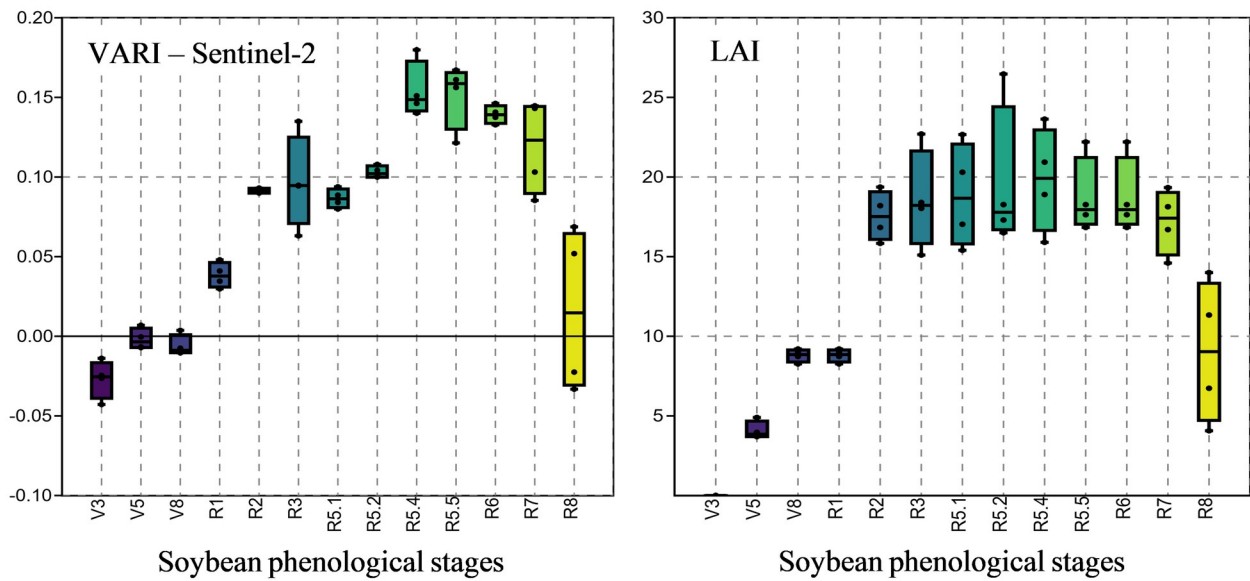

**Fig 10. Box plot for the VARI vegetation index, from the Sentinel-2 satellite, and for the soybean.** (LAI) leaf area index.

Furthermore, due to the pattern of variation in VI throughout the soybean development cycle, studies involving non-linear statistical modeling must be carried out in order to refine the identification of phenological stages via VI.

## Conclusions

Vegetation indices, obtained from orbital sensors, are effective in identifying soybean phenological stages relatively quickly and cheaply.

The RGB and NIR vegetation indices can be used to precisely identify some key phenological stages for soybean crop management, such as flowering ($R_1$ and $R_2$), end of pod development ($R_4$), the stage that characterizes the development of grains inside the pods ($R_{5.1}$) and stage that marks the physiological maturity of the crop ($R_8$).

No vegetation index, obtained by the Amazônia-1 and Sentinel-2 sensor systems, was 100% effective in identifying all phenological stages of the soybean crop.

The study suggests that the difference in spatial resolution of the two sensors evaluated, 10 meters per pixel of Sentinel-2 and 65 meters per pixel of Amazônia-1, is not a determining factor in the correct identification of soybean phenological stages.

## Author Contributions

**Conceptualization:** Airton Andrade da Silva, Francisco Charles dos Santos Silva, Ibrahim A. Saleh, Mohamed A. El-Tayeb, Alan Mario Zuffo.

**Data curation:** Airton Andrade da Silva, Francisco Charles dos Santos Silva, José Francisco da Crus Neto, Alan Mario Zuffo.

**Formal analysis:** Airton Andrade da Silva, Francisco Charles dos Santos Silva, Claudinei Martins Guimarães, Ibrahim A. Saleh, José Francisco da Crus Neto, Mohamed A. El-Tayeb, Mostafa A. Abdel-Maksoud, Jorge González Aguilera, Hamada AbdElgawad, Alan Mario Zuffo.

**Funding acquisition:** Francisco Charles dos Santos Silva, Claudinei Martins Guimarães, Ibrahim A. Saleh, Mohamed A. El-Tayeb, Mostafa A. Abdel-Maksoud, Hamada AbdElgawad, Alan Mario Zuffo.

**Investigation:** Airton Andrade da Silva, Francisco Charles dos Santos Silva, Claudinei Martins Guimarães, Ibrahim A. Saleh, José Francisco da Crus Neto, Mohamed A. El-Tayeb, Mostafa A. Abdel-Maksoud, Jorge González Aguilera, Hamada AbdElgawad, Alan Mario Zuffo.

**Methodology:** Airton Andrade da Silva, Francisco Charles dos Santos Silva, Claudinei Martins Guimarães, Ibrahim A. Saleh, José Francisco da Crus Neto, Mohamed A. El-Tayeb, Mostafa A. Abdel-Maksoud, Jorge González Aguilera, Hamada AbdElgawad, Alan Mario Zuffo.

**Project administration:** Francisco Charles dos Santos Silva, Claudinei Martins Guimarães, Alan Mario Zuffo.

**Resources:** Airton Andrade da Silva, Francisco Charles dos Santos Silva, Ibrahim A. Saleh, José Francisco da Crus Neto, Mohamed A. El-Tayeb, Mostafa A. Abdel-Maksoud, Hamada AbdElgawad, Alan Mario Zuffo.

**Software:** Airton Andrade da Silva, Francisco Charles dos Santos Silva, José Francisco da Crus Neto, Mohamed A. El-Tayeb, Mostafa A. Abdel-Maksoud, Jorge González Aguilera, Hamada AbdElgawad, Alan Mario Zuffo.

**Supervision:** Francisco Charles dos Santos Silva, Claudinei Martins Guimarães, Alan Mario Zuffo.

**Validation:** Airton Andrade da Silva, Francisco Charles dos Santos Silva, Claudinei Martins Guimarães, Ibrahim A. Saleh, José Francisco da Crus Neto, Mostafa A. Abdel-Maksoud, Jorge González Aguilera, Hamada AbdElgawad, Alan Mario Zuffo.

**Visualization:** Airton Andrade da Silva, Francisco Charles dos Santos Silva, Claudinei Martins Guimarães, Ibrahim A. Saleh, José Francisco da Crus Neto, Mohamed A. El-Tayeb, Mostafa A. Abdel-Maksoud, Jorge González Aguilera, Hamada AbdElgawad, Alan Mario Zuffo.

**Writing – original draft:** Airton Andrade da Silva, Francisco Charles dos Santos Silva, Claudinei Martins Guimarães, Ibrahim A. Saleh, José Francisco da Crus Neto, Mohamed A. El-Tayeb, Mostafa A. Abdel-Maksoud, Jorge González Aguilera, Hamada AbdElgawad, Alan Mario Zuffo.

**Writing – review & editing:** Airton Andrade da Silva, Francisco Charles dos Santos Silva, Claudinei Martins Guimarães, Ibrahim A. Saleh, José Francisco da Crus Neto, Mohamed A. El-Tayeb, Mostafa A. Abdel-Maksoud, Jorge González Aguilera, Hamada AbdElgawad, Alan Mario Zuffo.

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
