## [Decision Letter · Decision Letter 0]

28 Feb 2024

PONE-D-24-03115Spectral indexes with different spatial resolutions in recogniz-ing soybean phenologyPLOS ONE

Dear Dr. Silva,

Thank you for submitting your manuscript to PLOS ONE. After careful consideration, we feel that it has merit but does not fully meet PLOS ONE’s publication criteria as it currently stands. Therefore, we invite you to submit a revised version of the manuscript that addresses the points raised during the review process.

The manuscript must be corrected in all points indicated by the reviewers, such as:

1. The goal of the study is not clear. What the key research questions that your work address?

2. The methodology is too shallow and no allow to identify the novelty or contribution of the work. It was difficult to connect flowchart in Figure 1 with the method description.

3. It was very difficult to see whether the work effectively addressed the aim of the study in the conclusion.

4. To include some statistical findings in your abstract.

5. Performing exploratory data analysis to understand the variability within vegetation indexes across different phenological stages could be beneficial.

6. Verify the assumptions underlying Anderson's discriminant analysis, such as multivariate normality and homogeneity of covariance matrices, through diagnostic tests or graphical methods.

7. Evaluate the efficiency of vegetation indexes, compare the performance of the discriminant analysis model with baseline classifiers such as logistic regression or support vector machines.

8. Conduct sensitivity analysis to examine the impact of different parameters or settings on the results of Anderson's discriminant analysis.

We look forward to receiving your revised manuscript.

Kind regards,

Claudionor Ribeiro da Silva

Academic Editor

PLOS ONE

2. During your revisions, please confirm whether the wording in the title is correct and update it in the manuscript file and online submission information if needed.

“The authors extend their appreciation to the Researchers Supporting Project number (RSPD2024R678) King Saud University, Riyadh, Saud Arabia. This research was fi-nanced by a grant from the Foundationto Support Research and Scientific and Technological Development of Maranhão – FAPEMA”

“The authors extend their appreciation to the Researchers Supporting Project number (RSPD2024R678) King Saud University, Riyadh, Saud Arabia. The authors thank to the reviewers for their help in improving our manuscript, to Universidade Estadual do Maranhão (Balsas) for the technical, scientific, and structural support, and to CAPES for funding the research.”

“The authors extend their appreciation to the Researchers Supporting Project number (RSPD2024R678) King Saud University, Riyadh, Saud Arabia. This research was fi-nanced by a grant from the Foundationto Support Research and Scientific and Technological Development of Maranhão – FAPEMA”

7. PLOS requires an ORCID iD for the corresponding author in Editorial Manager on papers submitted after December 6th, 2016. Please ensure that you have an ORCID iD and that it is validated in Editorial Manager. To do this, go to ‘Update my Information’ (in the upper left-hand corner of the main menu), and click on the Fetch/Validate link next to the ORCID field. This will take you to the ORCID site and allow you to create a new iD or authenticate a pre-existing iD in Editorial Manager. Please see the following video for instructions on linking an ORCID iD to your Editorial Manager account: https://www.youtube.com/watch?v=_xcclfuvtxQ

8. We note that Figure 1 in your submission contain [map/satellite] images which may be copyrighted. All PLOS content is published under the Creative Commons Attribution License (CC BY 4.0), which means that the manuscript, images, and Supporting Information files will be freely available online, and any third party is permitted to access, download, copy, distribute, and use these materials in any way, even commercially, with proper attribution. For these reasons, we cannot publish previously copyrighted maps or satellite images created using proprietary data, such as Google software (Google Maps, Street View, and Earth). For more information, see our copyright guidelines: http://journals.plos.org/plosone/s/licenses-and-copyright.

Reviewers' comments:

Reviewer's Responses to Questions

**Comments to the Author**

1. Is the manuscript technically sound, and do the data support the conclusions?

Reviewer #1: Partly

Reviewer #2: Partly

Reviewer #3: Yes

2. Has the statistical analysis been performed appropriately and rigorously? 

Reviewer #1: Yes

Reviewer #2: Yes

Reviewer #3: Yes

3. Have the authors made all data underlying the findings in their manuscript fully available?

Reviewer #1: Yes

Reviewer #2: Yes

Reviewer #3: Yes

4. Is the manuscript presented in an intelligible fashion and written in standard English?

Reviewer #1: No

Reviewer #2: Yes

Reviewer #3: Yes

5. Review Comments to the Author

Reviewer #1: Major concern

1. The goal of the study is not clear to mean, or have not been well articulated in the last paragraph of the introduction. In fact, what the key research questions that your work seek to address?

2. The methodology is too shallow and has no debt to allow readers identify the novelty or contribution of the work. So much effort was spent on describing the data used, even in the aspect that talked about “Conducting the experiment.” Readers are expecting to see a step-by-step description of how the experiment was conducted. It was difficult to connect flowchart in Figure 1 with the method description. Every aspect of the work is just mixed up with content that ought to be in different section. In this current form, the work lack is no logical coherence. For example, I see you have two figures with caption as Figure 1. Secondly, for the second Figure 1 (flowchart), how was the reprojection done for both satellite images? How was the field data corroborated with the satellite image using the Anderson Discriminant Analysis? Lastly, what do you mean by Conclusions? You flowchart is supposed to be a concise summary of the approach adopted. But here, you confuse readers with “Conclusions”. How are we supposed to understand that while reading your methodology? Please, streamline the methodology to allow for easy reproducibility.

3. While reading your conclusion, it was very difficult to tell if the work actually addressed the goal of the study. I suggest you overhaul the entire manuscript, and then, clearly state the goal or objectives of the study. Then, try to show us in the conclusion that the goal has been achieved.

Minor comment

1. Why did you hyphenate “recogniz-ing” in the title (line 2)?

2. The title of this manuscript can be improved. Think about it carefully.

3. What do you mean by “Search Location” in line 100? Note that "Search location" and "Study location" are not exactly the same.

4. It would be very nice to include some statistical findings in your abstract.

Reviewer #2: Title: Spectral indexes with different spatial resolutions in recogniz-ing soybean phenology.

The title is not correct. "Indices" is the plural form of "index" when used in the context of measurements or indicators. Secondly the use of the hyphen in ‘recogniz-ing’ is unnecessary in this context. Similar mistake is also repeated in the Abstract section too, like: ‘cul-tivation’, and ‘demon-strates’. Therefore, the accurate title for the paper would be: "Spectral Indices with Different Spatial Resolutions in Recognizing Soybean Phenology." This usage adheres to standard scientific terminology in the field of remote sensing and geographic information systems.

In the abstract section, the findings are verbally expressed without substantiating the same with data. This is not a standard practice. The authors should have briefly described with data (in the abstract section) how different indices are correlated in identifying phonological stages of the soybean crop.

The spatial resolution of Amazonia 1 and sentinel 2 for (NIR, RGB) are 60 m and 10 m respectively. For a better accuracy assessment, the efficiency of each spectral indices needs to be checked at the same spatial resolution for both Amazonia 1 and sentinel 2 imageries, using resampling technique. This can be done either by upscaling the 10 m to 60 m or downscaling the 60 m to 10 m resolution. The authors have ignored this fact.

The temporal resolution i.e. the revisit frequency of each single SENTINEL-2 satellite is 10 days while the combined constellation revisit time is 5 days. The authors seem to have confused the revisit time of Sentinel 2 constellation with the temporal variation of MSI sensor onboard Sentinel-2. This is a major correction and it will impact the findings of this study.

Reviewer #3: Dear Authors,

Thank you for choosing PLOS for your interesting study. However, here are my specific suggestions and comments:

1. Consideration of Additional Statistical Tests: While Anderson's discriminant analysis is valuable for assessing classification accuracy, incorporating additional statistical tests such as ANOVA or pairwise comparisons could provide further insights into the significance of differences between vegetation indexes and phenological stages. This would strengthen the statistical robustness of your findings.

2. Exploratory Data Analysis for Variability: Prior to conducting discriminant analysis, performing exploratory data analysis to understand the variability within vegetation indexes across different phenological stages could be beneficial. Box plots or histograms could help visualize the distribution of index values and identify any outliers or trends that may impact the analysis.

3. Assessment of Model Assumptions: Verify the assumptions underlying Anderson's discriminant analysis, such as multivariate normality and homogeneity of covariance matrices, through diagnostic tests or graphical methods. Addressing violations of these assumptions ensures the reliability of the classification results.

4. Validation Techniques for Model Performance: Consider employing cross-validation or bootstrap resampling techniques to validate the performance of the discriminant analysis model. This would assess the generalizability of the classification results and provide confidence in the effectiveness of the selected vegetation indexes for phenological stage identification.

5. Comparison with Baseline Models: In addition to evaluating the efficiency of vegetation indexes, compare the performance of the discriminant analysis model with baseline classifiers such as logistic regression or support vector machines. This comparative analysis would offer a broader perspective on the suitability of different statistical approaches for phenological stage classification.

6. Sensitivity Analysis for Model Parameters: Conduct sensitivity analysis to examine the impact of different parameters or settings on the results of Anderson's discriminant analysis. This analysis would help identify optimal parameter choices and enhance the reproducibility of the classification outcomes.

By incorporating these suggestions, you can enhance the rigor and validity of your statistical analysis, providing a more comprehensive assessment of the efficiency of vegetation indexes in distinguishing soybean phenological stages.

Thank you in advance for taking into consideration my commnts and suggestions.

Kind regards,

Reviewer

6. PLOS authors have the option to publish the peer review history of their article (what does this mean?). If published, this will include your full peer review and any attached files.

Reviewer #1: No

Reviewer #2: **Yes: **Dr. A Salim Khan

Reviewer #3: No

---

## [Author Response · Author response to Decision Letter 0]

1 May 2024

Response to Reviewers

Regarding Figure 1, the satellite image obtained from Google Earth was replaced by an image collected by the CBERS4A satellite, a platform belonging to the Brazilian State that, like the LANDSAT system, has free distribution without copyright restrictions.

Reviewer #1: 

1. The goal of the study is not clear to mean, or have not been well articulated in the last paragraph of the introduction. In fact, what the key research questions that your work seek to address?

The aim of the research was to evaluate the efficiency of different vegetation indexes, obtained from satellites with different spatial resolutions, in discriminating the phenological stages of soybean crops. We made the necessary changes to the original material to meet the reviewer's recommendations.

2. The methodology is too shallow and has no debt to allow readers identify the novelty or contribution of the work. So much effort was spent on describing the data used, even in the aspect that talked about “Conducting the experiment.” Readers are expecting to see a step-by-step description of how the experiment was conducted. It was difficult to connect flowchart in Figure 1 with the method description. Every aspect of the work is just mixed up with content that ought to be in different section. In this current form, the work lack is no logical coherence. For example, I see you have two figures with caption as Figure 1. Secondly, for the second Figure 1 (flowchart), how was the reprojection done for both satellite images? How was the field data corroborated with the satellite image using the Anderson Discriminant Analysis? Lastly, what do you mean by Conclusions? You flowchart is supposed to be a concise summary of the approach adopted. But here, you confuse readers with “Conclusions”. How are we supposed to understand that while reading your methodology? Please, streamline the methodology to allow for easy reproducibility.

After the review, figure 1 now corresponds to the study location and the methodology flowchart becomes figure 4.

The reprojection of images was only necessary for Sentinel-2 images that are originally available in the WGS 84 coordinate reference system, requiring reprojection to Datum Sirgas 2000 UTM Zone 23S, which corresponds to the UTM projection zone of the study area.

Data collected in the field were correlated with satellite images, both representing the same sampling moment, and were used to verify the possibility of identifying a certain phenological stage, without the need to go to the field. To investigate this possibility, Anderson's Discriminant Analysis was applied to the satellite images and, after corrections, the Neural Networks technique was also applied.

The term “Conclusions” in the image referred to the study conclusions, but, in reviewing the image, the term was replaced by an icon that represents the interpretation of the obtained results.

All corrections, suggested by the reviewer for the methodology, were accepted.

3. While reading your conclusion, it was very difficult to tell if the work actually addressed the goal of the study. I suggest you overhaul the entire manuscript, and then, clearly state the goal or objectives of the study. Then, try to show us in the conclusion that the goal has been achieved.

The conclusions were rewritten to be well adjusted to the objective, as suggested by the reviewer.

Minor comment

1. Why did you hyphenate “recogniz-ing” in the title (line 2)?

Hyphen removed from the title

2. The title of this manuscript can be improved. Think about it carefully.

Suggestion accepted.

3. What do you mean by “Search Location” in line 100? Note that "Search location" and "Study location" are not exactly the same.

The terms have been standardized.

4. It would be very nice to include some statistical findings in your abstract.

Suggestion accepted.

Reviewer #2: 

1. Title: Spectral indexes with different spatial resolutions in recogniz-ing soybean phenology. The title is not correct. "Indices" is the plural form of "index" when used in the context of measurements or indicators. Secondly the use of the hyphen in ‘recogniz-ing’ is unnecessary in this context. Similar mistake is also repeated in the Abstract section too, like: ‘cul-tivation’, and ‘demon-strates’. Therefore, the accurate title for the paper would be: "Spectral Indices with Different Spatial Resolutions in Recognizing Soybean Phenology." This usage adheres to standard scientific terminology in the field of remote sensing and geographic information systems.

The reviewer's suggestion was fully met. 

2. In the abstract section, the findings are verbally expressed without substantiating the same with data. This is not a standard practice. The authors should have briefly described with data (in the abstract section) how different indices are correlated in identifying phonological stages of the soybean crop.

The reviewer's suggestion was fully met. 

3. The spatial resolution of Amazonia 1 and sentinel 2 for (NIR, RGB) are 60 m and 10 m respectively. For a better accuracy assessment, the efficiency of each spectral indices needs to be checked at the same spatial resolution for both Amazonia 1 and sentinel 2 imageries, using resampling technique. This can be done either by upscaling the 10 m to 60 m or downscaling the 60 m to 10 m resolution. The authors have ignored this fact.

In this work, the authors chose not to apply the resampling technique in order to simulate the real conditions of data use. So, when a user uses satellite image data, products are generated from images with their original spatial resolution, without resampling. Thus, it is possible to infer the influence of spatial resolution on the accuracy of vegetation indexes in identifying soybean phenological stages.

4. The temporal resolution i.e. the revisit frequency of each single SENTINEL-2 satellite is 10 days while the combined constellation revisit time is 5 days. The authors seem to have confused the revisit time of Sentinel 2 constellation with the temporal variation of MSI sensor onboard Sentinel-2. This is a major correction and it will impact the findings of this study.

The reviewer's suggestion was fully met. 

Reviewer #3: 

1. Consideration of Additional Statistical Tests: While Anderson's discriminant analysis is valuable for assessing classification accuracy, incorporating additional statistical tests such as ANOVA or pairwise comparisons could provide further insights into the significance of differences between vegetation indexes and phenological stages. This would strengthen the statistical robustness of your findings.

ANOVA was performed to identify differences between vegetation indexes at different phenological stages. However, the authors decided to present only the significance of its test (F Test) in figure 5. It was decided not to perform a pairwise comparison test, and, to represent the index values difference in the different phenological stages, it was chosen the line graph shown in figure 5.

2. Exploratory Data Analysis for Variability: Prior to conducting discriminant analysis, performing exploratory data analysis to understand the variability within vegetation indexes across different phenological stages could be beneficial. Box plots or histograms could help visualize the distribution of index values and identify any outliers or trends that may impact the analysis.

Before the ANOVA, box plot graphical analysis was performed to identify outliers, which were eliminated when identified. However, it was decided not to present such graphics in order to make the reading of the work more objective and fluid.

3. Assessment of Model Assumptions: Verify the assumptions underlying Anderson's discriminant analysis, such as multivariate normality and homogeneity of covariance matrices, through diagnostic tests or graphical methods. Addressing violations of these assumptions ensures the reliability of the classification results.

The reviewer's suggestion was fully met. 

4. Validation Techniques for Model Performance: Consider employing cross-validation or bootstrap resampling techniques to validate the performance of the discriminant analysis model. This would assess the generalizability of the classification results and provide confidence in the effectiveness of the selected vegetation indexes for phenological stage identification.

The cross-validation technique was applied to the Neural Network Analysis that was added to the study.

5. Comparison with Baseline Models: In addition to evaluating the efficiency of vegetation indexes, compare the performance of the discriminant analysis model with baseline classifiers such as logistic regression or support vector machines. This comparative analysis would offer a broader perspective on the suitability of different statistical approaches for phenological stage classification.

The reviewer's suggestion was fully met and the data was also subjected to Neural Network Analysis. This technique is capable of identifying non-linear patterns in the data set and thus maximizing the identification of groups of samples.

6. Sensitivity Analysis for Model Parameters: Conduct sensitivity analysis to examine the impact of different parameters or settings on the results of Anderson's discriminant analysis. This analysis would help identify optimal parameter choices and enhance the reproducibility of the classification outcomes. By incorporating these suggestions, you can enhance the rigor and validity of your statistical analysis, providing a more comprehensive assessment of the efficiency of vegetation indexes in distinguishing soybean phenological stages.

Specifically, for the suggestion of carrying out sensitivity analyzes to examine the impact of different parameters or configurations on the results of Anderson's discriminant analysis, the data were subjected to quadratic discriminant analysis, however this presented lower representation than the linear one, present in the article. Therefore, the authors decided not to present these results in the manuscript.

---

## [Decision Letter · Decision Letter 1]

4 Jun 2024

Spectral indices with different spatial resolutions in recognizing soybean phenology

PONE-D-24-03115R1

Dear Dr. Silva,

We’re pleased to inform you that your manuscript has been judged scientifically suitable for publication and will be formally accepted for publication once it meets all outstanding technical requirements.

Kind regards,

Claudionor Ribeiro da Silva

Academic Editor

PLOS ONE

Additional Editor Comments (optional):

Reviewers' comments:

Reviewer's Responses to Questions

**Comments to the Author**

1. If the authors have adequately addressed your comments raised in a previous round of review and you feel that this manuscript is now acceptable for publication, you may indicate that here to bypass the “Comments to the Author” section, enter your conflict of interest statement in the “Confidential to Editor” section, and submit your "Accept" recommendation.

Reviewer #1: All comments have been addressed

Reviewer #2: All comments have been addressed

2. Is the manuscript technically sound, and do the data support the conclusions?

Reviewer #1: Yes

Reviewer #2: Yes

3. Has the statistical analysis been performed appropriately and rigorously? 

Reviewer #1: Yes

Reviewer #2: Yes

4. Have the authors made all data underlying the findings in their manuscript fully available?

Reviewer #1: No

Reviewer #2: Yes

5. Is the manuscript presented in an intelligible fashion and written in standard English?

Reviewer #1: Yes

Reviewer #2: Yes

6. Review Comments to the Author

Reviewer #1: "Differentiation"?

In table 3 and other places, is it not better to just say, "Normalized Difference Vegetation Index"?

I think you should deal with this word: "Differentiation"

Reviewer #2: The comments have been addressed adequately. The manuscript now appears to be an actual contribution to science. I recommend that this manuscript may be accepted for publication.

7. PLOS authors have the option to publish the peer review history of their article (what does this mean?). If published, this will include your full peer review and any attached files.

Reviewer #1: No

Reviewer #2: **Yes: **Dr. A Salim Khan

---

## [Editor Report · Acceptance letter]

20 Aug 2024

PONE-D-24-03115R1 

PLOS ONE

Dear Dr. Silva, 

I'm pleased to inform you that your manuscript has been deemed suitable for publication in PLOS ONE. Congratulations! Your manuscript is now being handed over to our production team.

Kind regards, 

on behalf of

Dr. Claudionor Ribeiro da Silva 

Academic Editor

PLOS ONE